# Comparative Genomics and Functional Studies of Putative m^6^A Methyltransferase (METTL) Genes in Cotton

**DOI:** 10.3390/ijms232214111

**Published:** 2022-11-15

**Authors:** Junfeng Cao, Chaochen Huang, Jun’e Liu, Chenyi Li, Xia Liu, Zishou Zheng, Lipan Hou, Jinquan Huang, Lingjian Wang, Yugao Zhang, Xiaoxia Shangguan, Zhiwen Chen

**Affiliations:** 1Hainan Yazhou Bay Seed Laboratory, Sanya 572025, China; 2National Key Laboratory of Plant Molecular Genetics, Institute of Plant Physiology and Ecology/CAS Center for Excellence in Molecular Plant Sciences, Chinese Academy of Sciences, Shanghai 200032, China; 3Plant Biotechnology Research Center, Fudan-SJTU-Nottingham Plant Biotechnology R&D Center, Key Laboratory of Urban Agriculture (South), Ministry of Agriculture, School of Agriculture and Biology, Shanghai Jiao Tong University, Shanghai 200240, China; 4School of Life Sciences, Peking University, Beijing 100871, China; 5Esquel Group, 25 Harbour Road, Wanchai, Hong Kong, China; 6Institute of Cotton Research, Shanxi Agricultural University, Yuncheng 044099, China

**Keywords:** m^6^A methyltransferase, cotton, phylogenetics, VIGS, divergent function

## Abstract

N6-methyladenosine (m^6^A) RNA modification plays important regulatory roles in plant development and adapting to the environment, which requires methyltransferases to achieve the methylation process. However, there has been no research regarding m^6^A RNA methyltransferases in cotton. Here, a systematic analysis of the m^6^A methyltransferase (METTL) gene family was performed on twelve cotton species, resulting in six METTLs identified in five allotetraploid cottons, respectively, and three to four METTLs in the seven diploid species. Phylogenetic analysis of protein-coding sequences revealed that *METTL* genes from cottons, *Arabidopsis thaliana*, and *Homo sapiens* could be classified into three clades (METTL3, METTL14, and METTL-like clades). *Cis*-element analysis predicated the possible functions of *METTL* genes in *G. hirsutum*. RNA-seq data revealed that *GhMETTL14* (*GH_A07G0817*/*GH*_*D07G0819*) and *GhMETTL3* (*GH*_*A12G2586*/*GH*_*D12G2605*) had high expressions in root, stem, leaf, torus, petal, stamen, pistil, and calycle tissues. *GhMETTL14* also had the highest expression in 20 and 25 dpa fiber cells, implying a potential role at the cell wall thickening stage. Suppressing *GhMETTL3* and *GhMETTL14* by VIGS caused growth arrest and even death in *G. hirsutum*, along with decreased m^6^A abundance from the leaf tissues of VIGS plants. Overexpression of *GhMETTL3* and *GhMETTL14* produced distinct differentially expressed genes (DEGs) in *A. thaliana*, indicating their possible divergent functions after gene duplication. Overall, GhMETTLs play indispensable but divergent roles during the growth of cotton plants, which provides the basis for the systematic investigation of m^6^A in subsequent studies to improve the agronomic traits in cotton.

## 1. Introduction

N6-methyladenosine (m^6^A) methylation is the most abundant post-transcriptional modification of nucleotides in mRNA [1,2]. m^6^A RNA methylation is a dynamic process and highly conserved in mammals, flies, plants, etc., that is mediated by m^6^A writer, eraser, and reader proteins [3]. The m^6^A methylation writer complex is composed of the m^6^A methyltransferases and other factors, e.g., WTAP. METTL3 was the first m^6^A RNA methyltransferase identified in *Homo sapiens* [4]. Subsequently, METTL14 was proposed as the second m^6^A methyltransferase in humans with high homology to METTL3 [5,6,7]. Actually, subsequent studies showed that METTL14 had no methyltransferase activity, but could bind to RNA substrates by interacting with METTL3 to form a heterodimer [8,9,10]. WTAP was also discovered to be a major binding partner of the complex [11]. WTAP interacted with METTL3 and METTL14, and was required for METTL3–METTL14 heterodimer localization into the nucleus for the catalytic activity of the m^6^A methyltransferase complex [11].

Recent studies on m^6^A writers have also been conducted in plants extensively, including the m^6^A methyltransferase components being characterized in *Arabidopsis*. For example, AtMTA (METTL3 human homolog protein) is one of the earliest discovered m^6^A methyltransferases in *Arabidopsis* [12]. AtMTA proteins were actively distributed in the reproduction organs and apical meristems of *Arabidopsis* plants. AtMTA protein was found to be required for m^6^A mRNA methylation, and its function disruption prevented plant embryo development, which further resulted in embryo lethality [12]. In addition, PtrMTA, a *Populus trichocarpa* methyltransferase MTA homologous protein, was colocalized in the nucleus. PtrMTA-overexpressing plants significantly enhanced their tolerance to drought stress by increasing the density of trichomes and roots in poplar [13]. AtMTB is the human METTL14 homologous protein identified in *Arabidopsis*, but its function is unknown [14]. However, ClMTB, an AtMTB homolog protein in watermelon, and its expression was induced under drought stress. Overexpression of *ClMTB* enhanced drought tolerance by elevating the reactive oxygen species (ROS) scavenging ability and alleviating photosynthesis inhibition in tobacco [15]. Both these MTAs and MTBs belong to the MT-A70-like (MT) proteins [3]. These results imply that the m^6^A methyltransferases might serve as the positive regulatory factors in response to the different adverse stresses.

As m^6^A methyltransferases are highly conserved, the protein methyltransferases in *H. sapiens* and *Arabidopsis* would provide the necessary sequence references to identify the orthologous proteins or genes in other plant species. Cotton (*Gossypium*) is the most important fiber crop worldwide, and this genus contains more than 50 species [16,17,18,19]. There are many publicly available genomes for *Gossypium*, such as diploid D sub-genome species *Gossypium thurberi* (D_1_), *G. raimondii* (D_5_), *G. turneri* (D_10_) [20,21,22,23]; diploid A sub-genome species *G. herbaceum* (A_1_), *G. arboreum* (A_2_) [24,25,26]; diploid F sub-genome species *G. longicalyx* (F_1_) [27]; diploid G sub-genome species *G. bickii* (G_1_) [28], *G. australe* (G) [29]; and allotetraploid species *G. hirsutum* (AD_1_), *G. barbadense* (AD_2_), *G. tomentosum* (AD_3_), *G. mustelinum* (AD_4_), and *G. darwinii* (AD_5_) [26,30,31,32,33,34,35,36,37]; as well as *Gossypium* sister genera *Gossypioides kirkii* [38]. These genome sequences provide a good platform for performing comparative genomics analysis and dissecting their gene functions. As the m^6^A methyltransferase (METTL) genes in *Gossypium* have not been investigated, several representative *Gossypium* species were selected to characterize the m^6^A methyltransferase genes and their roles in cotton.

In this study, we performed genome-wide screening for METTL genes in cotton, based on data gathered from recent whole-genome sequencing efforts. We used an in silico approach to identify and characterize *METTL* genes, and then focused on the characterization and phylogenetic relationships of *Gossypium METTLs*. The *METTL* gene structures, conserved domains, as well as cis-elements were explored in this systematical analysis. Moreover, the tissue-specific expression patterns and the transcriptional responses of *GhMETTLs* to abiotic stresses were examined. The possible biological functions of *GhMETTL3* and *GhMETTL14* were investigated in *G. hirsutum* and *A. thaliana* using the VIGS method and overexpression of transformation, respectively. Our data indicate the conservation of m^6^A methyltransferase (METTL) genes in cotton, and provide evidence for the divergent roles of *GhMETTL3* and *GhMETTL14* during *Gossypium* evolution.

## 2. Results

### 2.1. Identification and Chromosomal Location of RNA Methyltransferase Genes in G. hirsutum

Six methyltransferase-like genes (*METTL*) were identified in the *G. hirsutum* genome using Hmmersearch with conserved domains (PF05063 for mRNA: m^6^A methyltransferase) (Table 1). These 6 *GhMETTL* genes were dispersed over 6 of the 26 *G. hirsutum* chromosomes, with all homoeologs conserved (Figure 1). Among them, *GH_A12G2586/GH_D12G2605* (*GhMETTL3_A/D*) were the homologs of *METTL3*, and *GH_A07G0817/GH_D07G0819* (*GhMETTL14_A/D*) were the homologs of *METTL14* in *H. sapiens*. *GH_A06G0112/GH_D06G0096* were another two copies of methyltransferase-like genes in *G. hirsutum* (Figure 1).

### 2.2. Structural Organization of GhMETTL Genes

Less than a threefold variation in length was detected in the predicted coding sequences (CDS) for these *GhMETTL*s, from 1269 bp for *GH*_*A06G0112*/*GH_D06G0096* to 3558 bp for *GH_A07G0817* (Table 1), which translates to proteins ranging from 422 amino acids (aa) to 1185 aa. Predicted isoelectric points (pI) for members of this family range from 5.92 to 7.18 (Table 1). All putative *GhMETTL* genes contain introns (Appendix A), which exhibit considerable variations in length and number. In general, homoeologous *GhMETTL* genes show highly similar intron patterns; however, intron structure among homoeologous pairs can exhibit variation in intron number (5 to 8) and length (Appendix A). One of the homoeologous gene pairs did exhibit divergence in intron length, namely, the fourth intron of *GH_D12G2605* (*GhMETTL3_D*) was significantly less than that of *GH_A12G2586* (*GhMETTL3_A*). Characterizing the gene structure in the diploid parental species for the homoeologs suggests that the length variation of the fourth intron occurred before the divergence of the allotetraploid (Appendix A). In addition, the phylogenetic relationship of the *GhMETTL* gene family was consistent with the intron/exon structure characterized (Appendix A). 

### 2.3. Phylogenetic Analysis of METTL Genes in Gossypium

We further evaluated the general preservation of METTL genes in 12 *Gossypium* species, *Gossypioides kirkii*, *Arabidopsis thaliana*, and *Homo sapiens* (Table 2). Two METTL genes (METTL3 and METTL14) were identified in *H. sapiens*. In *A. thaliana*, three METTL genes (AtMTA, AtMTB, and AtMTC) were identified; the relative of *Gossypium*, *Gossypioides kirkii*, had three METTL genes identified. The 12 *Gossypium* species surveyed recovered a minimum of three putative METTL genes in diploid species and six in five allotetraploid species (Table 2). Among D genome species, three METTL gene were identified in *G. thurberi* (D_1_), *G. raimondii* (D_5_), and *G. turneri* (D_10_), respectively. The two cultivated diploid species of *G. herbaceum* (A_1_) and *G. arboreum* (A_2_) had four copies. The sister species of A-genome *G. longicalyx* (F_1_) contained three METTL genes. METTL copy numbers in the allotetraploid species surveyed were conserved in six METTL genes in five species: *G. hirsutum* (AD_1_), *G. barbadense* (AD_2_), *G. tomentosum* (AD_3_), *G. mustelinum* (AD_4_), *G. darwinii* (AD_5_) (Table 2). Notably, this high copy number in the allotetraploids is almost double the copy number in the diploids, likely reflective of the duplicated history of cotton. Comparatively, the METTL copy number is generally stable after polyploidization. The diploid A genome cotton species included here both appeared to have undergone a homoeolog gain (three versus four). 

To further investigate the evolutionary relationships between the METTL genes provided here, we specifically assessed these relationships using the protein sequences of 61 METTL genes, including 53 *Gossypium*, 3 *Gossypioides kirkii*, 3 *A. thaliana*, and 2 *H. sapiens* METTL genes for phylogenetic analysis (Figure 2). Three clades (METTL3, METTL14, and METTL-like) were robustly supported in *Gossypium*, with one *A. thaliana* or *Gossypioides kirkii* gene associated with each clade, respectively, but shared METTL3 and METTL14 clades with *H. sapiens.*

Overall, the expected diploid–polyploid topology is reflected in the tree for each set of orthologous/homoeologous genes, indicating general preservation during diploid divergence and through polyploid evolution. That is, the number of METTL genes in allotetraploids was generally additive with respect to the model diploid progenitors, with each homoeolog (A_t_ or D_t_) sister to their respective parental copies. METTL3 and METTL14 clades had equal numbers of METTL genes (Figure 2). In clades METTL3 and METTL14, genes related to *AtMTA* and *AtMTB* were exhibited parallelly, and there was no duplication in *Gossypium* species, compared to *A. thaliana* and *H. sapiens*. Therefore, it is speculated that the function of METTL genes in these clades of cotton is similar to that of the corresponding METTL genes in *Arabidopsis* and *H. sapiens*.

### 2.4. Cis-Element Analysis of METTL Genes in G. Hirsutum

*Cis*-elements are responsive to corresponding stimulations to regulate the expression of genes [41]. In this study, a 1.5 kb upstream region from the start codon of each METTL gene in *G. hirsutum* was extracted to investigate putative *cis*-elements involved in the mediation of gene expression using the PlantCARE server [42]. We completely identified 213 *cis*-elements among 6 *GhMETTL* genes, ranging from 24 in *GH_A12G2586* to 44 in *GH_D06G0096* (Appendix A). All *GhMETTL* gene promoters had several TATA-box and CAAT-box elements (Appendix A). Four *GhMETTL* gene promoters processed at least one abscisic acid responsiveness element (ABRE), and three *GhMETTL* genes had at least one AE-box (part of a module for light response in *A. thaliana*). *GhMETTL14_A* (*GH_A07G0817*) processed the element involved in HD-Zip 1 (involved in differentiation of the palisade mesophyll cells in *A. thaliana*). Six *GhMETTLs* had light responsiveness elements (GA-motif, GT1-motif, and G-box) and at least two MYB or one MYC binding site element (MYB/MYC). In addition, MYB binding sites involved in drought inducibility (MBS) also existed in these *GhMETTLs* genes.

### 2.5. Expression Patterns of GhMETTL Genes in Different G. hirsutum Tissues

The expression patterns of a gene family can provide valuable information to predict the possible biological functions of each gene. Analysis of six *GhMETTL* genes showed that most genes have different spatial expression patterns. For instance, the expression levels of *GhMETTL14* (*GH_A07G0817*/*GH_D07G0819*) and *GhMETTL3* (*GH_A12G2586*/*GH_D12G2605*) in the root, stem, leaf, torus, petal, stamen, pistil, and calycle were significantly higher than the other two *GhMETTL*-like (*GH_A06G0112*/*GH_D06G0096*) genes (Figure 3A). *GhMETTL14* (*GH_A07G0817*/*GH_D07G0819*) and *GhMETTL3* (*GH_A12G2586*/*GH_D12G2605*) presented the highest expression level in the pistil than other tissues (Figure 3A). In addition, *GhMETTL14* (*GH_A07G0817*/*GH_D07G0819*) still showed significantly higher expression in seed and root samples at different time points after seed germination (Figure 3B). In ovule samples at different development stages, *GhMETTL14* (*GH_A07G0817*/*GH_D07G0819*) had the highest expression levels, followed by *GhMETTL3* (*GH_A12G2586*/*GH_D12G2605*) (Figure 3C). In different development periods of fiber samples, *GhMETTL14* (*GH_A07G0817*/*GH_D07G0819*) had the highest expression in 20 and 25 dpa fiber cells (Figure 3D), suggesting that this gene might play an important role at the cell wall thickening stage. *GhMETTL3* (*GH_A12G2586*/*GH_D12G2605*) also presented higher expression in 20 and 25 dpa fiber cells than in 5 and 10 dpa cells (Figure 3D). The above two genes were the homologs of the *H. sapiens METTL3* and *METTL14*, respectively (Figure 2). These results suggest that *GhMETTL* genes may not be involved in the regulation of fiber cell elongation (Figure 3D). *GhMETTL3* and *GhMETTL14* might serve as the functional genes in charge of cotton fiber cell wall development. Based on the gene expression data, two *GhMETTL*-like (*GH_A06G0112*/*GH_D06G0096*) genes were barely expressed across different tissues, which might diverge to become pseudogenes.

### 2.6. Expression Changes of GhMETTL Genes in G. Hirsutum under Different Stresses

A variety of abiotic stresses pose threats to the growth and development of cotton plants. Therefore, we comprehensively analyzed the expression changes of *GhMETTL* genes under simulated drought (PEG 6000), salt (NaCl), heat, and cold abiotic stresses using RNA-seq data (Appendix A). At different time points of PEG6000-simulated drought conditions, the expression levels of six *GhMETTL* genes were slightly decreased, but the difference was not significant (|lg_2_ (fold change)|≥1| as the threshold of differentially expressed genes). Among them, the *GhMETTL14_A* (*GH_A07G0817*) gene was downregulated after 1 h PEG treatment (Appendix A), but not changed at the time points of 3, 6, and 12 h. The expression of other *GhMETTL* genes did not change after PEG treatment for 1, 3, 6, and 12 h (Appendix A). These results indicate that few *GhMETTL* genes were involved in response to drought stress in *G. hirsutum*. Meanwhile, at the four time points of salt or high-temperature stress, the expressions of six *GhMETTL* genes both did not change after 1, 3, 6, and 12 h of NaCl or heat treatment (Appendix A). 

As for the four time points of low-temperature stress in *G. hirsutum*, *GhMETTL14* (*GH_A07G0817*/*GH_D07G0819*) were upregulated after 6 h of low-temperature treatment (Appendix A). However, the expressions of *GhMETTL3* (*GH_A12G2586*/*GH_D12G2605*) were repressed after 12 h low-temperature treatment. These results indicate their divergent regulating roles under low-temperature stress. The expression levels of these genes did not change at other time points. The overall results show that *GhMETTL3* and *GhMETTL14* played an important role in response to the adverse stress in cotton, and it might be worth further exploring their biological functions. 

### 2.7. Suppressing GhMETTL3 and GhMETTL14 Caused Growth Arrest in G. hirsutum

To explore the role of the identified *GhMETTLs* during the growth of cotton plants, the virus-induced gene silencing (VIGS) approach was used to suppress the expression of *GhMETTL3* and *GhMETTL14* (Figure 4). About 400 bp fragments from the conserved regions of both genes were inserted into the tobacco rattle virus (TRV) vector, respectively, and TRV-vector-carrying Agrobacterium strains were then co-inoculated into 1-week-old cotton plant cotyledons by the needleless syringe method. From those silenced cotton plants, we selected each independent VIGS line for phenotypic analyses (Figure 4A–C). The qPCR results show that the expression levels of *GhMETTL3* and *GhMETTL14* genes in silenced plants were significantly downregulated (Figure 4D,E). Along with the decreased expression of the *GhMETTL* genes, suppressing *GhMETTL3* and *GhMETTL14* both caused growth arrest in *G. hirsutum,* with a more severe withered phenotype produced by *GhMETTL14* (Figure 4A). The resulting phenotypes also indicate that the inhibition of *GhMETTL14* resulted in slight yellowing of the leaves (Figure 4B). One month later, downregulation of *GhMETTL3* and *GhMETTL14* by VIGS led to the death of cotton plants (Figure 4C). Then, the m^6^A abundances in the leaf tissues of *GhMETTL3* and *GhMETTL14* VIGS plants were determined by quantitative mass spectrometry (MS) analysis (Figure 4F). For the TVR2 control leaf tissue, the m^6^A/A ratio of total RNA was approximately 0.075% (Figure 4F), which was significantly higher than that in *GhMETTL3*-silenced and *GhMETTL14*-silenced plants, ranging from 0.050% to 0.057% of m^6^A/A ratio, respectively. To provide more evidence of the molecular functions of *GhMETTLs*, we checked the expression levels of growth-related genes whose mRNA contains m^6^A according to the public cotton m^6^A-seq data [31]. The results show that the expression level of *MYB* (*Gohir.D06G213100*), *bHLH* (*Gohir*.*A11G121700*), and epidermal growth factor receptor (*Gohir*.*A01G103800*) decreased both in the *GhMETTL3* as well as the *GhMETTL14* VIGS plants (Appendix A). Furthermore, 10 μL actinomycin D was used for the VIGS plants to inhibit the transcription, and the mRNA stability of the three genes was reduced dramatically (Appendix A). Overall, the results show that the decreased m^6^A abundance might cause the growth arrest of *G. hirsutum* juvenile plants.

### 2.8. Subcellular Localization of GhMETTL3 and GhMETTL14 Proteins

To predict the subcellular localization of GhMETTL3 and GhMETTL14 proteins, we constructed two fusion vectors. A C-terminal fusion to the GFP gene was generated, driven by the CaMV 35S promoter, to determine the subcellular localization. Then, GhMETTL3:GFP and GhMETTL14:GFP fusion proteins were transiently expressed in tobacco epidermal cells and viewed using a confocal fluorescence microscope (Figure 5). The transient expression of GhMETTL3:GFP and GhMETTL14:GFP fusion proteins were specifically localized in the nucleus, not other subcellular compartments. Combined with the m^6^A ratio in the VIGS cotton, these results indicate that the GhMETTL3 and GhMETTL14 proteins act as the writers of m^6^A in cotton.

### 2.9. Overexpression of GhMETTL3 and GhMETTL14 Produced Distinct Differentially Expressed Genes (DEGs) in A. Thaliana

To further analyze the biological function of *GhMETTL3 and GhMETTL14*, we transformed these two genes into *Arabidopsis* using the floral dip method, and then we obtained the T_1_ generation transgenic plants. However, we did not observe any significant phenotypic difference between two transgenic and wild type *Arabidopsis* plants during the whole growth period. To gain insight into the molecular mechanisms by overexpressing *GhMETTL3 and GhMETTL14* into *Arabidopsis* plants, we performed RNA-seq of leaf tissues from two transgenic plants compared with the wild type.

Transcriptome data showed that there indeed existed differentially expressed transcripts in the leaf samples after transformation. In total, we identified 71 differentially expressed genes (DEGs) in *GhMETTL3* overexpressed *Arabidopsis* plants compared with the wild type, among which 42 were induced and 29 were repressed (Appendix A, Appendix A). Meanwhile, 1274 DEGs were identified in *GhMETTL14* overexpressed *Arabidopsis* plants with 315 upregulating and 958 downregulating (Appendix A, Appendix A). 

Subsequently, these DEGs were subjected to KEGG pathway analysis to identify the functional categorization. Appendix A lists the results of the KEGG analysis for DEGs. DEGs between *OEGhMETTL3* and WT were assigned to 17 KEGG pathways, including RNA degradation, phenylpropanoid biosynthesis, ascorbate and aldarate metabolism, inositol phosphate metabolism, protein processing in endoplasmic reticulum, nitrogen metabolism, and the MAPK signaling pathway in plants, etc. (Appendix A). In addition, DEGs between *OEGhMETTL14* and WT were significantly enriched in 93 KEGG pathways, mainly including phenylpropanoid biosynthesis, glutathione metabolism, protein processing in endoplasmic reticulum, ascorbate and aldarate metabolism, starch and sucrose metabolism, pentose and glucuronate interconversions, pyruvate metabolism, fatty acid elongation, fatty acid biosynthesis, fatty acid degradation, nitrogen metabolism, RNA degradation, as well as the MAPK signaling pathway in plants (Appendix A). The above results reveal that these two methyltransferase genes (*GhMETTL3* and *GhMETTL14*) exerted extensive and distinct effects on the life processes in *A. thaliana* plants.

### 2.10. GhMETTL3 and GhMETTL14 Exhibited Divergent Functions in Transgenic Arabidopsis 

Although no significant phenotypic changes were observed after these two genes were transformed into *A. thaliana*, transcriptome data still proved that *GhMETTL3* and *GhMETTL14* exhibited functional differentiation in transgenic *Arabidopsis*. We selected a total of 88 genes that were differentially expressed between transgenic and wild type *Arabidopsis*, and these DEGs could be classified into 11 families, including ethylene responsive transcription factor gene, bHLH transcription factor genes, WRKY transcription factor genes, dirigent protein genes, cytochrome P450 genes, calcium-binding protein genes, ABC transporter genes, glutathione S-transferase genes, detoxification genes, xyloglucan hydrolase genes, and CCR4-associated factor 1 homolog genes. Among them, six gene families comprising nine ABC transporter genes (Figure 6A), thirteen cytochrome P450 genes (CYP) (Figure 6B), six detoxification genes (DTX) (Figure 6C), four bHLH transcription factor genes (Figure 6D), three dirigent protein genes (DIR) (Figure 6E), and nine glutathione S-transferase genes (Figure 6F) were all downregulated in *OEGhMETTL14* transgenic *Arabidopsis* plants, but unchanged in *OEGhMETTL3* compared with the wild type plants. These results indicate that these two genes were functionally inconsistent.

In addition, among thirteen calcium-binding protein (CML) genes (Appendix A), the expression levels of eight *CML* genes (*CML12*, *CML27*, *CML39*, *CML41*, *CML42*, *CML44*, *CML46*, and *CML47*) were downregulated in *OEGhMETTL14*, but not changed in *OEGhMETTL3* transgenic *Arabidopsis* plants compared with the wild type. Four *CML* genes (*CML23*, *CML37*, *CML38*, and *CML40*) were repressed in *OEGhMETTL14*, but induced in *OEGhMETTLL3* transgenic *Arabidopsis* plants. Expression of one *CML* gene (*CML45*) was increased in *OEGhMETTL3*, but unchanged in *OEGhMETTL14* transgenic *Arabidopsis* plants (Appendix A). For another fifteen ethylene responsive transcription factor genes (ERF) (Appendix A), there were six ERF genes (*ERF019*, *ERF043*, *ERF060*, *ERF061*, *ERF114*, and *ERF115*) being downregulated in *OEGhMETTL14* plants compared with the wild type, but not changed in *OEGhMETTL3*. Meanwhile, three ERF genes (*ERF1*, *ERF13*, and *ERF1B*) were reduced in *OEGhMETTL14*, but elevated in *OEGhMETTL3* plants. There still existed six ERF genes (*ERF5*, *ERF016*, *ERF017*, *ERF018*, *ERF104*, and *ERF105*) that were upregulated in *OEGhMETTL3* plants, but unchanged in *OEGhMETTL14* (Appendix A). In addition, the presence of divergence between these two *GhMETTL* genes was also revealed from the expression patterns of twelve WRKY transcription factor genes (Appendix A). Thereinto, the expressions of eight WRKY genes (*WRKY8*, *WRKY28*, *WRKY29*, *WRKY48*, *WRKY51*, *WRKY54*, *WRKY62*, and *WRKY70*) were downregulated in *OEGhMETTL14* compared with the wild type, but not changed in *OEGhMETTL3*. Four WRKY genes (*WRKY33*, *WRKY40*, *WRKY46*, and *WRKY53*) presented reduced expressions in *OEGhMETTL14*, but induced in *OEGhMETTL3* (Appendix A). Expressions of three CCR4-associated factor 1 homolog genes (*CAF1-11*, *CAF1*-*5*, and *CAF1-9*) were upregulated in *OEGhMETTL3* plants, but not changed in *OEGhMETTL14* (Appendix A). Finally, discordant expression patterns of four xyloglucan hydrolase genes (XTH) were also observed between *OEGhMETTL14* and *OEGhMETTL3* transgenic *Arabidopsis* plants (Appendix A). Among them, the expressions of two *XTH* genes were upregulated in *OEGhMETTL3* plants compared with that of wild type, but not changed in *OEGhMETTL14*. Two XTH genes exhibited repressed expressions in *OEGhMETTL14*, but upregulated in *OEGhMETTL3* compared with the wild type (Appendix A). The above results further confirm that the incongruent expression levels of these DEGs were caused by the transformations of *GhMETTL3* and *GhMETTL14* into *Arabidopsis thaliana*. Their biological functions might also be diverged and is worth further exploring in subsequent studies.

## 3. Discussion

N6-methyladenosine (m^6^A) in RNA represents an indispensable post-transcriptional modification in gene regulation. It was involved in RNA stability and degradation [43] and mRNA alternative splicing [44], thus affecting translation efficiency [45] and miRNA processing [46]. Meanwhile, the m^6^A modification plays a critical role in plant growth, development, and response to the adverse environmental stresses. The m^6^A writer components include MTA, MTB, FIP37, VIRILIZER, and HAKAI in *A. thaliana* [3,47,48,49]. When the expression level of m^6^A writer components, such as m^6^A methyltransferase, were reduced, the abundance of m^6^A also showed a dramatic reduction [12]. Knockout of m^6^A methyltransferase MTA prevented development and eventually led to embryonic lethality, while inactivation of other writer components resulted in increasing trichome branch phenotypes, as well as the over-proliferation of shoot apical meristem [3,47,48].

Previous reports have indicated that m^6^A modification is involved in abiotic and biotic stress responses [50,51,52,53]. The increase in 5’ UTR m^6^A level would promote protein translation in response to the heat shock stress [54]. Another study showed that m^6^A methylation was involved in salt stress tolerance in *Arabidopsis*. Both mutants of m^6^A writer components, such as MTA and MTB, displayed salt-sensitive phenotypes along with a reduced m^6^A level [55]. On the other hand, m^6^A methylation level showed a reduction after exposure to virus infection in *Nicotiana tabacum* and drought stress in *Zea mays*, respectively [56,57]. In addition, the expression of m^6^A methyltransferase *ClMTB* in watermelon was also induced by drought stress. Overexpression of *ClMTB* in tobacco plants increased their drought tolerance by enhancing ROS scavenging activity [15]. Overexpressing *Populus trichocarpa* m^6^A methyltransferase also enhanced the tolerance to drought stress by increasing the trichome and root density in poplar [13]. In our study, we also found the expression of *GhMETTL14_A* (*GH_A07G0817*) gene was repressed after simulated drought PEG treatment. These results indicate that METTL genes that mediated m^6^A modification were involved in response to environmental stresses in plants.

For the emergence of homologous genes in a single gene family after duplication, homologous genes generally have different fates: both genes retained for dosage effect or diverged to produce new function [58,59]. In METTL3 mutants, the translation of mRNAs containing m^6^A modifications in the 5’UTR was reduced, indicating translation efficiency was influenced by 5’UTR m^6^A in cells [60,61]. Early reports also found that m^6^A modifications in 5’UTR regions were positively correlated with gene expression in *Arabidopsis* [62]. These investigations suggested that the METTL3 protein was the most active m^6^A methyltransferase enzyme to catalyze m^6^A RNA methylation, and a core m^6^A methyltransferase component. However, studies found that METTL14 had no methyltransferase activity, but could form a stable antiparallel heterodimer with METTL3 to further bind to the RNA substrates in cells [8,9,10]. In *Arabidopsis*, the AtMTA protein was necessary for m^6^A mRNA methylation, and loss of function led to plant embryo development arrest, and further to lethality [12]. However, the AtMTB identified might have no m^6^A methyltransferase activity in *Arabidopsis* [14]. Furthermore, only one METTL3 or MTA homologous protein was identified in barley and *Micromonas pusilla*. In these two species, the METTL14 or MTB protein may have undergone functional gene loss during evolution or evolved into a distinct molecular role of a new gene family, but not the MT-A70-like protein [3]. In our study, the similar phenomenon of GhMETTL gene’s function divergence was observed in *Gossypium*. For instance, the expressions of *GhMETTL14* (*GH_A07G0817*/*GH_D07G0819*) were induced after low-temperature treatment, but *GhMETTL3* (*GH_A12G2586*/*GH_D12G2605*) were repressed under low-temperature stress. Meanwhile, *GhMETTL3* and *GhMETTL14* also exhibited divergent functions in transgenic *Arabidopsis* based on the transcriptome data. Compared to the wild type, the expressions of DEGs between *OEGhMETTL3* and *OEGhMETTL14* transgenic *Arabidopsis* were always inconsistent. These results imply that the homologous genes in a single gene family, such as m^6^A methyltransferases METTL3 and METTL14, have divergent regulatory roles during the process of growth and development of plants and animals.

Although there have been many studies on the response of METTL in regulating plant growth and development, as well as responses to adverse stresses in plants, there are few studies on the exploration the potential roles of *METTLs* in cotton. The overall results of this study show that *GhMETTL3* and *GhMETTL14* play different roles in cotton; it would be worth further exploring their biological functions with modern biotechnologies, such as gene editing.

## 4. Materials and Methods

### 4.1. Identification of METTL Family Genes and METTL Proteins in Diploid and Tetraploid Gossypium Species

The genome sequences of cotton species were downloaded from the CottonGen database (https://www.cottongen.org/ (accessed on 1 April 2020)) [63], including *G. raimondii* [20], *G. herbaceum* [26], *G. arboreum* [24], *G. hirsutum* [30], *G. barbadense*, *G. tomentosum*, *G. mustelinum*, and *G. darwinii* [26,30,31,32,33,34,35,36,37]. To identify all putative METTL proteins in each genome assembly, the METTL protein conserved domains (PF05063 for MTA70 superfamily) were used to develop a hidden Markov model [64] profile matrix via the hmmbuild program from the HMMER 3.3.1 package [65] with default parameters. This HMM profile matrix was used in conjunction with hmmersearch with default parameters against these *Gossypium* genome databases to identify putative METTL genes (*GhMETTLs*) [66,67]. Previously identified METTL gene sequences from *A. thaliana* (*AtMTA: AT4G10760.1* and *AtMTB: AT4G09980.1*) and *H. sapiens* (METTL, NM_019852.4, and NM_020961.3) were retrieved from the TAIR database [40] and NCBI database (https://www.ncbi.nlm.nih.gov) for phylogenetic comparison, respectively. The presence of conserved domains in each *Arabidopsis* and *Gossypium* gene was verified by the SMART version 9 conserved domain search tool [68] and Pfam 35.0 databases [69]. 

### 4.2. Chromosomal Location and Gene Structure Analyses 

Chromosomal locations for the identified *GhMETTLs* were extracted from the genome annotation gff3 file [30,36]. Chromosomal locations of the predicted *GhMETTLs* were visualized using TBtools windows-x64_1_098769 [70], and the exon–intron structure of each gene was displayed using the online tool GSDS 2.0 [71]. The number of amino acids, molecular weight (MW), and theoretical isoelectric point (pI) of putative *GhMETTL* proteins were determined using the ProtParam tool [72]. The GFF3 file was included as Appendix A named as “METTL-GFF3”.

### 4.3. Sequence Alignment and Phylogenetic Tree Construction

Complete amino acid sequences for *METTL* genes from 12 *Gossypium* species, *Gossypioides kirkii*, *A. thaliana*, and *H. sapiens* were aligned using MAFFT version 6 with the G-INS-i algorithm [73]. As METTLs are single-domain proteins with a conserved MTA70 domain, the NJ phylogenetic tree was constructed using MEGA version 11.0 [74] with p-distance model and uniform rates by sampling 1000 bootstrap replicates. The alignments, and tree files were included as Appendix A named as “METTL-NJ3 and METTL-aa”, respectively.

### 4.4. Analysis of Cis-Acting Element in Promoter Regions of GhMETTLs

The upstream sequences (1.5 kb) [75] of the *GhMETTL* genes were retrieved from the *G. hirsutum* genome sequence based on the gene locations [30,36]. Then, the retrieved promoter sequences were submitted to PlantCARE [42] to identify the potential *Cis*-acting element. 

### 4.5. Expression Patterns of GhMETTLs in Different Tissues and Stress Conditions

Raw RNA-seq data for *G. hirsutum* seed, root, stem, leaf, torus, petal, stamen, ovary, calyx, ovule (-3 dpa, -1 dpa, 0 dpa, 1 dpa, 3 dpa, 5 dpa, 10 dpa, 20 dpa, 25 dpa, 35 dpa) and fiber (5 dpa, 10 dpa, 20 dpa, 25 dpa) were downloaded from the NCBI Sequence Read Archive (PRJNA 248163) [30,36], represented by one library each. Reads were mapped to the *G. hirsutum* genome [36] via HISAT2 2.2.1 software with default parameters, and read abundance was calculated via StringTie v2.2.0 [76,77]. Read counts were normalized in R 3.2 using the RUVSeq package version 3.0 [78] and the internal control reference gene *GhUBQ7*, which is detected at relatively constant levels across different cotton samples [79]. Potential batch effects were corrected by the ComBat-seq sva v3.36.0 [80]. Gene expression was estimated by Ballgown [77], using fragments per kilobase million (FPKM) values to calculate the gene expression levels across libraries. Expression levels of *G. hirsutum* leaf RNA-seq data (in FPKM) for each *GhMETTL* gene under drought, salt, heat, and cold stress (time points: 0, 1, 3, 6, 12 h) were retrieved from the ccNET database [81]. Genes were considered differentially expressed if expression varied more than twofold with a *p*-value of less than 0.05. TBtools windows-x64_1_098769 [70] was used to display the gene expression patterns from the calculated FPKM values [82].

### 4.6. Virus-Induced Gene Silencing (VIGS) Assays

The virus-induced gene silencing (VIGS) assays were performed based on previous reports [83,84,85]. Briefly, to knockdown the expression of the GhMETTL genes, ~400 bp fragments of the *GhMETTL3* and *GhMETTL14* cDNA from upland cotton R15 were PCR-amplified using *TransTaq*^®^ DNA Polymerase High Fidelity (AP131-11, TransGen Biotech, Beijing, China) with specific primers (Appendix A). The resulting PCR products were infused to pTRV2 to produce the VIGS vectors, named pTRV2-*GhMETTL3* and pTRV2-*GhMETTL14*, respectively. The pTRV1 and pTRV2-*GhMETTL3* or pTRV2-*GhMETTL14* vectors were introduced into the Agrobacterium strain GV3101 through electroporation (Bio-Rad, Hercules, CA, USA). For the VIGS assay, the transformed Agrobacterium colonies containing pTRV1 and pTRV2-*GhMETTL3* or pTRV2-*GhMETTL14* were grown overnight at 28 °C. The Agrobacterium cells were collected and resuspended in the infiltration medium (10 mM MgCl_2_, 10 mM MES, and 200 mM acetosyringone) and subsequently adjusted to an OD600 of 0.5. The Agrobacterium strains containing the pTRV1 and pTRV2-*GhMETTL3* or pTRV2-*GhMETTL14* vectors were mixed at a ratio of 1:1. Cotton seedlings with mature cotyledons but without a visible rosette leaf (7 days after germination) were infiltrated by inserting the Agrobacterium suspension into the cotyledons via a syringe. The plants were then grown at 22 °C in pots arranged in a growth chamber under a 16 h light–8 h dark cycle and at a humidity of 60%. The mutant of *GhSU* has an obvious leaf-yellowing phenotype during the seeding stage, and we used *GhSU* as the positive control for VIGS. For mRNA stability assay, 10 μL actinomycin D was used to treat the VIGS plants as well as the negative control. After 1, 3, and 6 h, the treated leaves were collected to detect the expression levels.

### 4.7. RNA Extraction, cDNA Synthesis, and RT-qPCR Expression Analyses

These experiments were conducted with the methods reported previously [86,87,88]. In brief, total RNAs from cotton leaf were extracted using a RNAprep pure plant kit (TIANGEN, Shanghai, China). The resulting RNAs were treated with DNase I prior to synthesizing cDNA with *TransScript*^®^ First-Strand cDNA Synthesis SuperMix (AT301-02, TransGen Biotech, Beijing, China); these products were diluted fivefold before use. For quantitative real-time PCR, Primer v5.0 software was used to design gene-specific forward and reverse primers (Appendix A). Analyses were performed with SYBR-Green PCR Mastermix (TaKaRa) on a cycler (Mastercycler RealPlex; Eppendorf Ltd., Shanghai, China). The *G. hirsutum histone-3* (*GhHIS3*) genes were used as internal references, and the relative amount of amplified product was calculated following the 2^−∆∆Ct^ method [89].

### 4.8. Quantitative Analysis of m^6^A Level in Cotton Leaf Tissue

The quantification of m^6^A level in cotton leaf was conducted following the methods of a previous report [90]. In brief, mRNA was extracted from 10 μg total RNA for each sample with Dynabeads™ mRNA Purification Kit (Invitrogen, 61006). Then, 150 ng purified RNA was digested by 1 U nuclease P1 (Sigma, N8630) in 10 mM NH_4_Ac reaction buffer, made up to 20 μL, at 42 °C for 3 h. Subsequently, 1 U rSAP (NEB, M0371S) and 5 μL rCutSmart buffer were added to the mixture, and incubated at 37 °C for an additional 6 h. In addition, 1 μL digested RNA was injected into a LC-MS/MS (AB SCIEX QTRAP 5500) to detect the mononucleotide. The m^6^A level was detected and quantified in the positive ion multiple reaction monitoring (MRM) mode with nucleoside to base ion mass transitions (282.0 to 150.1 for m^6^A and 268.0 to 136.0 for A). The concentrations of m^6^A level in cotton leaf were calculated from the standard curve generated from pure nucleoside standards.

### 4.9. Subcellular Localization of GhMETTL3 and GhMETTL14 Proteins and Microscopic Observation

The protein subcellular localization experiments were carried out as described in a previous study [90]. The coding region of GhMETTL3 and GhMETTL14 was fused to GFP by *pEASY*^®^-Basic Seamless Cloning and Assembly Kit (CU201-02, TransGen Biotech, Beijing, China). The vectors carrying p35S::GhMETTL3-GFP and p35S::GhMETTL14-GFP were introduced into Agrobacterium cells, which were infiltrated with a syringe into the abaxial surface of *N. benthamiana* leaves for transient expression. Three days later, the infiltrated areas were observed under a confocal microscope (LSM510, Zeiss, Oberkochen, Germany). 

### 4.10. Arabidopsis Transformation 

The floral dip method was used for *Arabidopsis* transformation [91]. In brief, to overexpress *GhMETTL3* and *GhMETTL14* in *A. thaliana*, the full-length cDNA of *GH_A12G2586* (*GhMETTL3_A*) and *GH_A07G0817* (*GhMETTL14_A*) was cloned into the pCAMBIA2301 vector driven by the CaMV35S promoter, respectively. The recombinant plasmid was transferred into Agrobacterium cells and then transformed into *A. thaliana* plants via the floral dip method [91,92]. Kanamycin painting and PCR analysis confirmed the transgenic lines.

### 4.11. RNA Sequencing of OEGhMETTL3, OEGhMETTL14, and Wild Type A. Thaliana Plants 

Total RNA was isolated from the OEGhMETTL3, OEGhMETTL14, and wild type *A. thaliana* plants. Library construction and sequencing were performed following previous reports [93,94]. In brief, 1 μg of purified mRNA was selected for cDNA library construction. mRNA was purified from total RNA using poly-T oligo-attached magnetic beads. cDNA fragments 240 bp in length were preferentially selected, and the library fragments were purified using an AMPure XP system (Beckman Coulter, Beverly, USA). Finally, the PCR products were purified (AMPure XP system), and library quality was assessed on an Agilent Bioanalyzer 2100 system. After cluster generation, the library preparations were sequenced on an Illumina HiSeqTM 2500 platform, and 150 bp paired-end reads were generated. Three biological replicates were performed for the transgenic and wild type *A. thaliana* plants. RNA-sequencing data that support the findings of this study have been deposited in the Genome Sequence Archive in the BIG Data Center of Sciences (https://bigd.big.ac.cn/ (accessed on 28 September 2021)) under accession CRA008376.

### 4.12. Read Data Quality Control, Mapping, and Calculations of the Differentially Expressed Genes

Raw data (raw reads) in fastq format were first processed through FASTX-Toolkit (http://hannonlab.cshl.edu/fastx_toolkit/ (accessed on 2 February 2010)). Clean data (clean reads) were obtained by removing reads containing adapters, reads containing poly-N sequences, and low-quality reads from the raw data. The downstream analyses were based on clean high-quality data. RNA-seq data were mapped to the reference genomes using HISAT2 software [76,77]. Only reads with a perfect match or one mismatch were further analyzed to calculate the expression values. Differential expression analysis of the two groups was performed using DESeq2 (v1.0) [95]. The resulting *p*-values were adjusted using Benjamini and Hochberg’s approach for controlling the false discovery rate (FDR). Genes with an adjusted *p*-value < 0.01 and twofold change (up and down) were defined as differentially expressed. TBtools windows-x64_1_098769 [70] was used to display the gene expression patterns from the TPM values.

### 4.13. Gene Annotation and Enrichment Analyses

Differentially expressed gene functions were annotated based on the following databases: Nr (NCBI nonredundant protein sequences, ftp://ftp.ncbi.nih.gov/blast/db/ (accessed on 1 July 2008)), Nt (NCBI nonredundant nucleotide sequences, ftp://ftp.ncbi.nih.gov/blast/db/ (accessed on 1 July 2008)), and Kyoto Encyclopedia of Genes and Genomes (KEGG) (http://www.genome.jp/kegg/ (accessed on 1 January 2020)). KOBAS was used to test the statistical enrichment of the differentially expressed genes in the KEGG pathways [96].

## 5. Conclusions

We reported the structures, domains, phylogenetics, divergence, and *cis*-elements analyses results, systematically, of *Gossypium METTL* genes. We also investigated the expression patterns of METTL family genes in different cotton tissues and under different stress conditions to predict their possible biological functions. Both *GhMETTL3* and *GhMETTL14* genes were variably expressed in different cotton tissues, with particularly high expression in pistil and ovules. Suppressing *GhMETTL3* and *GhMETTL14* by VIGS led to development arrest in *G. hirsutum*. Overexpression of *GhMETTL3* and *GhMETTL14* into *Arabidopsis* did not produce significant phenotype differences, but effected the expressions of variant family genes, implying their divergent functions in the regulation of growth and development of cotton plants. Together, our results provide candidate genes to facilitate the functional identification of METTL genes in subsequent studies.

## Figures and Tables

**Figure 1 ijms-23-14111-f001:**
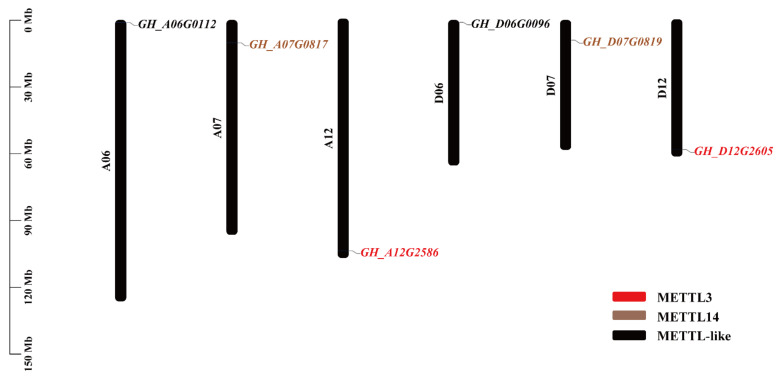
Dispersed distribution of METTL genes in *G. hirsutum* (AD1) chromosomes: 6 *GhMETTL* genes are scattered over 6 of the 26 *G. hirsutum* chromosomes.

**Figure 2 ijms-23-14111-f002:**
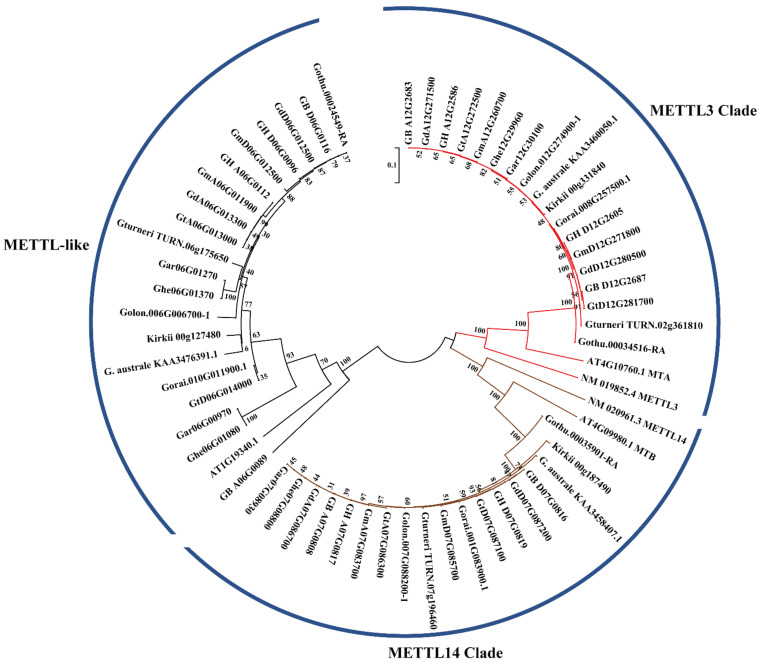
Phylogenetic analysis of *METTL* genes from 12 *Gossypium* species, *Gossypium kirkii*, *Arabidopsis thaliana*, and *Homo sapiens*. The phylogenetic tree was established with entire amino acid sequences using NJ methods. The numbers on the branches indicate bootstrap support values from 1000 replications.

**Figure 3 ijms-23-14111-f003:**
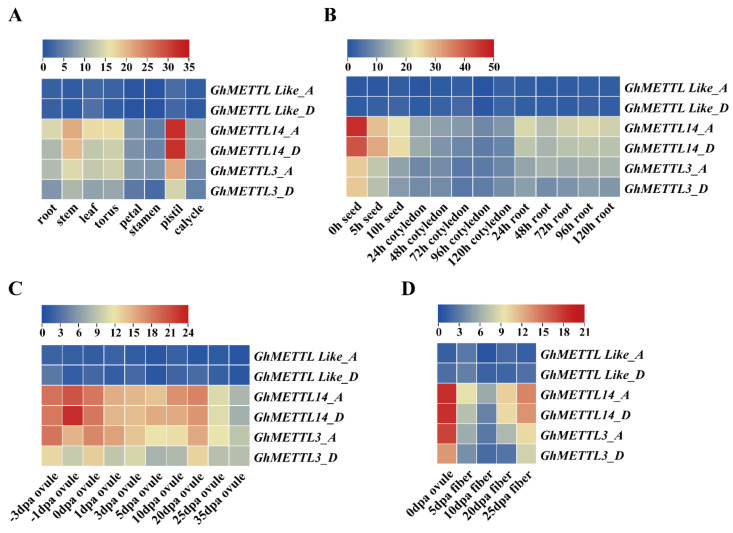
Expression patterns of *GhMETTL* genes in different cotton tissues and fiber cells of different stages based on the RPKM values of RNA-seq data. (**A**) Expression profiles of *GhMETTL* genes in eight cotton tissues. (**B**) Expression patterns of *GhMETTL* genes in seed germination, cotyledons, and roots after germination. (**C**) Expression patterns of *GhMETTL* genes in ovules of different stages. (**D**) Expression patterns of *GhMETTL* genes in fibers of different stages.

**Figure 4 ijms-23-14111-f004:**
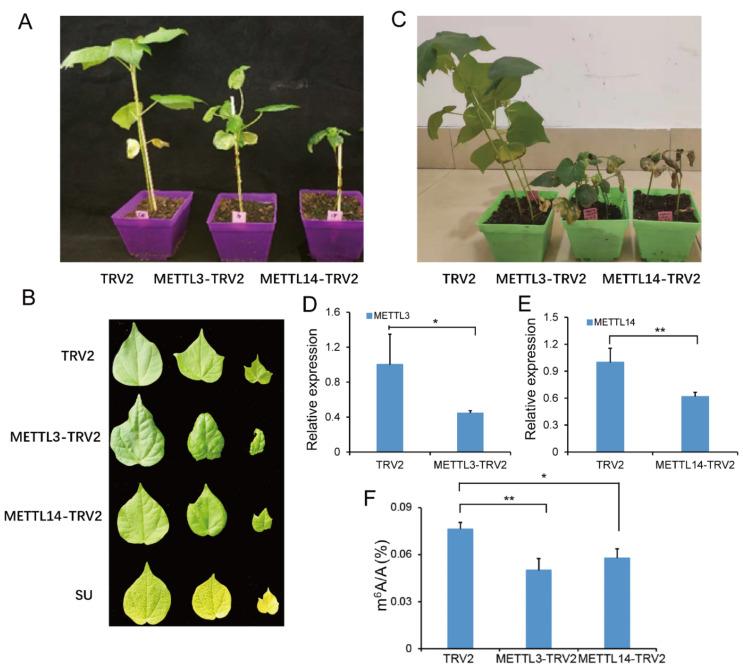
Suppressing *GhMETTL3* and *GhMETTL14* by virus-induced gene silencing (VIGS) approach. The 400 bp CDS fragments of two genes were inserted into the tobacco rattle virus (TRV) vector, and TRV-vectors-carrying Agrobacterium strains were then co-inoculated into 1-week-old cotton plant cotyledons by the needleless syringe method. (**A**) Suppressing *GhMETTL3* and *GhMETTL14* both caused development arrest in *G. hirsutum*. (**B**) The leaf phenotypes of *GhMETTL3* and *GhMETTL14* VIGS cotton plants compared with the TRV2 blank vector plants. (**C**) Downregulation of *GhMETTL3* and *GhMETTL14* by VIGS led to the death of cotton plants. (**D**,**E**) The qPCR data identified the repressed expression levels of *GhMETTL3* (**D**) and *GhMETTL14* (**E**) genes in silenced plants. (**F**) The decreased m^6^A abundance from the leaf tissues of *GhMETTL3* and *GhMETTL14* VIGS plants determined by quantitative mass spectrometry (MS) analysis compared with the TVR2 control. Means of triplicates ± SD, ** *p* < 0.01, * *p* < 0.05, Student’s *t*-test.

**Figure 5 ijms-23-14111-f005:**
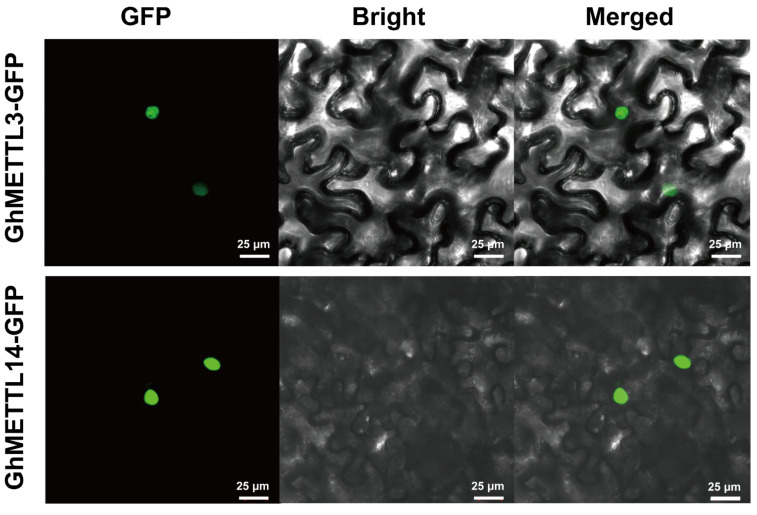
Subcellular localization assays of GhMETTL3 and GhMETTL14, showing nuclear localization of GhMETTL3-GFP (upper) and GhMETTL14-GFP (bottom). Agrobacterium cells harboring the fusion protein genes were infiltrated into leaves of *Nicotiana benthamiana* through the abaxial surface, and 72 h later the samples were observed under a confocal microscope (LSM510, Zeiss). Scale bar, 25 μm.

**Figure 6 ijms-23-14111-f006:**
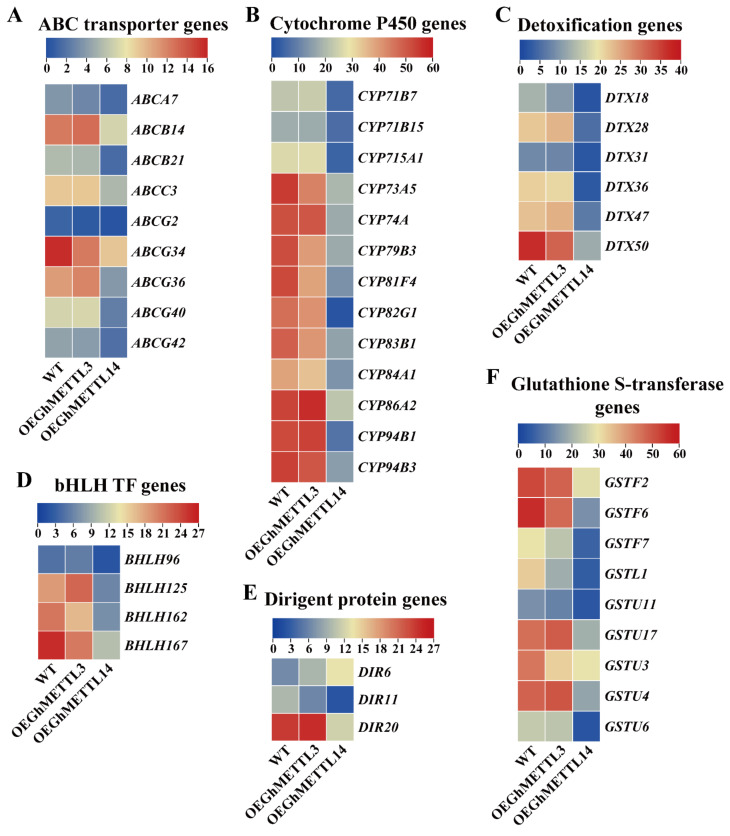
Expressions of six family genes were reduced in *OEGhMETTL14* but not changed in *OEGhMETTL3* transgenic *Arabidopsis* plants compared with the wild type. (**A**) ABC transporter genes. (**B**) cytochrome P450 genes (CYP). (**C**) detoxification genes (DTX). (**D**) bHLH transcription factor genes. (**E**) dirigent protein genes (DIR). (**F**) glutathione S-transferase genes.

**Table 1 ijms-23-14111-t001:** Sequence characteristics of *GhMETTL* (*G. hirsutum* methyltransferase) genes and proteins.

Gene Name	Locus Name	Chr	Genomics Position	CDS	No. of Introns	Size (aa)	MW	pI
*GhMETTL3_A*	*GH_A12G2586*	A12	104,206,704-104,211,250	2106	6	701	78.50	5.98
*GhMETTL3_D*	*GH_D12G2605*	D12	58,367,589-58,370,785	2106	6	701	78.42	5.92
*GhMETTL14_A*	*GH_A07G0817*	A07	10,130,789-10,135,646	3558	5	1185	132.59	7.10
*GhMETTL14_D*	*GH_D07G0819*	D07	8,927,821-8,932,500	3555	5	1184	132.49	6.92
*GhMETTL-Like_A*	*GH_A06G0112*	A06	1,007,934-1,011,291	1269	8	422	48.45	7.18
*GhMETTL-Like_D*	*GH_D06G0096*	D06	890,591-893,931	1269	8	422	48.47	6.88

Note: bp, base pair; Chr, chromosome; aa, amino acid; MW (estimated), molecular weight; kDa, kilodalton; pI, isoelectric point. The gene accessions were downloaded from the CottonGen database [39].

**Table 2 ijms-23-14111-t002:** Characteristics of the putative methyltransferase (METTL) genes in 12 *Gossypium* species, *Gossypium kirkii*, *Arabidopsis thaliana*, and *Homo sapiens*.

Species	METTL Number	METTL3 Homologs	METTL14 Homologs	METTL-Like Homologs
*Gossypium hirsutum*	6	GH_A12G2586/GH_D12G2605	GH_A07G0817/GH_D07G0819	GH_A06G0112/GH_D06G0096
*Gossypium barbadense*	6	GB_A12G2683/GB_D12G2687	GB_A07G0808/GB_D07G0816	GB_A06G0089/GB_D06G0116
*Gossypium tomentosum*	6	GtA12G272500/GtD12G281700	GtA07G086300/GtD07G087100	GtA06G013000/GtD06G014000
*Gossypium mustelinum*	6	GmA12G260700/GmD12G271800	GmA07G083700/GmD07G085700	GmA06G011900/GmD06G012500
*Gossypium darwinii*	6	GdA12G271500/GdD12G280500	GdA07G086700/GdD07G087200	GdA06G013300/GdD06G012500
*Gossypium herbaceum*	4	Ghe12G29960	Ghe07G08800	Ghe06G01080/Ghe06G01370
*Gossypium arboreum*	4	Gar12G30100	Gar07G08930	Gar06G00970/Gar06G01270
*Gossypium longicalyx*	3	Golon.012G274900-1	Golon.007G088200-1	Golon.006G006700-1
*Gossypium thurberi*	3	Gothu.00034516-RA	Gothu.00035901-RA	Gothu.00024549-RA
*Gossypium raimondii*	3	Gorai.008G257500.1	Gorai.001G083900.1	Gorai.010G011900.1
*Gossypium turneri*	3	Gossypium turneri_TURN.02g361810	Gossypium turneri_TURN.07g196460	Gossypium turneri_TURN.06g175650
*Gossypium australe*	3	G. australe_KAA3460050.1	G. australe_KAA3458407.1	G. australe_KAA3476391.1
*Gossypioides kirkii*	3	Kirkii_Version3_Juiced.00g331840.m01.polypeptide	Kirkii_Version3_Juiced.00g187490.m01.polypeptide	Kirkii_Version3_Juiced.00g127480.m01.polypeptide
*Arabidopsis thaliana*	3	AT4G10760.1_MTA	AT4G09980.1_MTB	AT1G19340.1
*Homo sapiens*	2	NM_019852.4_METTL3	NM_020961.3_METTL14	

Note: The gene accessions from 12 *Gossypium* species and *Gossypium kirkii* were downloaded from the CottonGen database [39]. The gene accessions of *Arabidopsis thaliana* and *Homo sapiens* were downloaded from the TAIR database [40] and NCBI database (https://www.ncbi.nlm.nih.gov (accessed on 1 July 2008)), respectively.

## Data Availability

RNA-sequencing data that support the findings of this study have been deposited in the Genome Sequence Archive in the BIG Data Center of Sciences (https://bigd.big.ac.cn) under accession CRA008376.

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
