# Peer review of "Comparative Genomics and Functional Studies of Putative m^6^A Methyltransferase (METTL) Genes in Cotton"

_ijms, 2022, doi:10.3390/ijms232214111_

Round 1

Reviewer 1 Report

N6-methyladenosine (N6mA/m6A) in RNA plays important regulatory roles in eukaryotic development. This function is mediated by RNA methyltransferases, particularly METTL gene family. In this report Cao et al., have undertaken a bioinformatic approach to identify METTL gene families in twelve cotton species. They have narrowed down their METTL genes to GhMETTL14 (GH_A07G0817/GH_D07G0819) and GhMETTL3 (GH_A12G2586/GH_D12G2605). They studied these gene expression in various organs in cotton plants. They also performed both suppression of the genes in vivo as well as overexpression to study global gene expression. Overall, the experiments are performed with proper control, and manuscript is written well. A detailed study of cotton METTL3/14. Some additional experiments are suggested to strengthen the manuscript.

Major issues:

1.     How does the authors know GhMETTL3/14 are catalytically active? Any biochemistry data to prove? I understand in figure 6F there is mass-spec data. Scrambled TRV2 control vs. MTTL3-TRV and MTTL3-TRV would have been ideal. I didn’t see that overexpression lines have any global mass-spec results. Indeed, RNA methylation is only 1% of same RNA max, thus overexpression can show an increased m6A/A. Similarly, the authors should perform catalytically inactive mutant (active site mutation) overexpression to show m6A/A level do not change in RNA.

2.     In figure 8B, Pajust values are low, please explain the context.

3.     How do we know that GhMETTL3/14 overexpression in Arabidopsis leads to certain gene overexpression or reduction? It could be due to increase half lives of RNA -species or rapid degradation due to additional m6A on those RNA. Can the author show some candidate RNA half-life studies.

Minor issues:

4.     Fig 8, 9 and 10 could be supplementary figure

5.     Table 1, MW (estimated) must be noted. Generally, post-translational modifications can change the mass of the protein.

Reviewer 2 Report

This a detailed study about methyltransferase proteins in an important crop (cotton).  The study include comparative genomics, annotation and characterization of homologous genes across various species in the genus, as well as expression data and functional experimental manipulation on the effects of the expression of  these genes (silencing and over expression).

The study is very complete and generally well written.

My main point of critique to the study are on the use of FPKM as for comparing expression levels across samples. It is well documented that this type of read count normalization is not suitable to compare abundance across samples. I suggest replacing or complementing these with alterbative normalization methods (DESeq2, or edgeR TMM are good options)

The phylogenetic tree inference could also be improved and their methods should be more detailed. In particular, there is no justification to default to NJ over model based inference (ML, and Bayesian methods are available and more appropiate). There is no mention on if the reading frame is preserved in the alignment in which case the use of codon models could be considered.

The version of each program mus be provided and the alignments, GFF3, and tree files should be included as supplementary files in the final version.

Below you'll find a few minor errors/suggestions.

The title of the manuscript is confusing. I suggest to simplify it e.g.: "Comparative genomics of m6A methyltransferase (METTL) genes reveal plant development role in cotton (/Gossypium hirsutum/, Malvaceae)"

|    Line | Comment                                                                                                                                                                                                                                                                                                                                                   |   |
|---------+-----------------------------------------------------------------------------------------------------------------------------------------------------------------------------------------------------------------------------------------------------------------------------------------------------------------------------------------------------------+---|
|      47 | replace "identified" with "proposed"                                                                                                                                                                                                                                                                                                                      |   |
|      54 | replace "Recently, researches"  with "Recent studies"                                                                                                                                                                                                                                                                                                     |   |
|      72 | italicize "H. sapiens"                                                                                                                                                                                                                                                                                                                                    |   |
|      74 | remove "genus" from (/Gossypium/ genus)                                                                                                                                                                                                                                                                                                                   |   |
|      75 | remove  "As much Gossypium plant genome information was publicly available"                                                                                                                                                                                                                                                                               |   |
|      75 | replace previos with "There are many publicly avalible genomes for /Gossypium/, suach as..."                                                                                                                                                                                                                                                              |   |
|      81 | remove "will" from "will provide"                                                                                                                                                                                                                                                                                                                         |   |
|      87 | replace "results" with "efforts"                                                                                                                                                                                                                                                                                                                          |   |
|     104 | confirm the term "homoeologs" is appropiate or refers to genaral homology.                                                                                                                                                                                                                                                                                |   |
|     457 | indicate version of HMMER                                                                                                                                                                                                                                                                                                                                 |   |
|  461-62 | provide accession numbers of reference sequences                                                                                                                                                                                                                                                                                                          |   |
|     464 | provide version of SMART and PFAM                                                                                                                                                                                                                                                                                                                         |   |
|     469 | provide version of TBtools                                                                                                                                                                                                                                                                                                                                |   |
|     472 | provide version of ProtParam                                                                                                                                                                                                                                                                                                                              |   |
|     475 | provide version for MAFFT                                                                                                                                                                                                                                                                                                                                 |   |
|     476 | Much information is needed for phylogenetic analyses, what type of sequence (DNA/AA)? Substitution model? Model selection? why NJ over other algorithms? this does not seem a trivial tree and better exploration of the gene family evolution is strongly recommended. Why use version 6 over current version is 11? Consider  other programs: eg IQTREE |   |
|     488 | provide version of HiSTA2                                                                                                                                                                                                                                                                                                                                 |   |
|     489 | provide version for StringTie                                                                                                                                                                                                                                                                                                                             |   |
|     489 | space between "R" and the version of the langiage interpeter "R"                                                                                                                                                                                                                                                                                          |   |
| 489-490 | "using the RUVseq package version #### [76]"                                                                                                                                                                                                                                                                                                              |   |
|     492 | remove "improved version of ComBat," or provide reference for ComBat                                                                                                                                                                                                                                                                                      |   |
|     492 | Provide version of Combat-Seq sva package version                                                                                                                                                                                                                                                                                                         |   |
|     493 | It is well documneted that FPKM nromalizations can induce errors in comparing gene expression. Use TMM instead or in addition to. (eg. see Wagner and  Lynch. Theory Biosci. 2012 doi: 10.1007/s12064-012-0162-3. Epub 2012 Aug 8. PMID: 22872506.)                                                                                                       |   |
|     498 | provide version for Tbtools                                                                                                                                                                                                                                                                                                                               |
|     525 | provide version of Primer5                                                                                                                                                                                                                                                                                                                                |   |
|     578 | provide version for DESeq2 packageUse DESEq normalization values inster                                                                                                                                                                                                                                                                                   |   |
|     581 | Use DESeq normalized counts instead of FPKM, see https://hbctraining.github.io/DGE_workshop/lessons/02_DGE_count_normalization.html                                                                                                                                                                                                                                                                                                           |   |

| Fig 2    | use italics for scientific names in graph                       |
| Fig 3    | Show an inset with untransformed branch lengths and scale. Provide full species names for terminals or color code them |
| Fig 5    | Previous comments on the use FPKM apply to the figure. These values are not comparable across samples |

Reviewer 3 Report

This is a meticulously performed, quite data-rich descriptive study bringing some novel insights into the molecular apparatus responsible for RNA methylation in plants. Unfortunately, not all author´s conclusions are adequately documented and there is quite a lot to improve with respect to presentation of the story, and I thus have to ask for a major revision (providing, at the same time, the authors an opportunity and suggestions for possible further improvement).

Below I am listing my main concerns.

1) Critical:

The main problem of this paper is insufficient documentation of the RNA-seq experiment described in Chaper 2.9. Although the raw data were adequately deposited in a public repository, the actual list of differentially expressed genes found in this study is nowhere to be found. This list must be included, at best as Supplementary data (probably an Excel file).

2) Study design, data presentation and interpretation issues:

- The title does not adequately capture contents of the paper, since the mutant deffect is very clearly not "developmental arrest" (in the sense of failure to perform a specific developmental step) but a simple growth/survival problem. I suggest renaming, e.g. to "Phylogenetic and functional studies of m6A methyltransferase (METTL) genes in cotton".

- The phylogenetic tree in Fig.3 should be presented as unrooted and in a form that preserves branch length (perhaps in "radiation" style?). The authors should also clearly state whether METTLs exhibit any variability in domain organization, because their method of phylogenyu inferrence is only good for sequence collections with uniform domain composition. If domain composition is variable, only domains found in all the studied genes should be taken into account.

- From the gene expression data, GH_A06G0112 and GH_D06G0096 (GhMETTLike_A and GhMETTLike_D) are barely, if at all, expressed. Could these be pseudogenes? The authors should comment on this.

- There is a lot of internal redundancy in the text and between the text and tables (see specific comments in the attached file).

- Data shown in Fig.2 would be more suitable for presentation in the form of a table rather than this figure.

3) Suggestions for possible further improvement:

The differential gene expression data documenting the effect of GhMETTL3 and GhMETTL14 overexpression come from a heterologous, possibly artifactual system that is very difficult to interpret. If the authors were able to show, by semiquantitative RT-PCR or by RT-qPCR, for at least a couple of differentially expressed genes (e.g. two to five) that their expression changes (presumably in the opposite direction) in the sickly VIGS plants, they would have a much stronger and more interesting story.

There are also numerous relatively mino, formal, language or style issues that I have commented upon in the attached file.

Round 2

Reviewer 1 Report

In my previous report I requested the authors to demonstrate GhMETTL3/14 are catalytically active (comment #1). The authors have replied “At present, we have already created GhMETTL3 overexpression and RNAi transgenic cotton materials to test the biochemical activity site and the content variations of m6A/A. We are also working on the other precious comments mentioned by the reviewer, and we feel that these results would be better presented in the next paper.”

I agree that a detailed biochemical and expression studies can be an additional manuscript. However, we are not sure if GhMETTL3/14 are m6A enzymes. The authors could either (1) change the title to “Comparative genomics and functional studies of putative m6A methyl- transferase (METTL) genes in cotton”. Or (2) include in the discussion sections that the catalytic activities of these ORFs are not determined and give persuasive explanation that these two ORFs are the most likely candidate. If there is no definitive experimental validation, the title is misleading.

My previous comment #3 was again a functional validation of m6A methyltransferase on RNA stability. Suppressing GhMETTL3 and GhMETTL14 by virus-induced gene silencing (VIGS) approach is established in authors laboratory, a few qPCR in control and suppression cell lines will be enough and was not a time-consuming request. The authors must do the science of validation to justify the work.

Reviewer 3 Report

The manuscript has undergone many generally positive changes and the majority of my concerns has been addressed adequately. I especially like the added information included in Table 2 that makes the paper easier to orient in, and the new organization of the supplements.

The only exception is the phylogenetic tree in Fig. 2, where a rooted distanceless tree has been replaced by an unrooted one but still stripped of branch length (phylogenetic distance) information  I therefore feel that another round of revision is necessary, namely:

The distanceless tree in Fig. 2  should be replaced by one that includes distance information (drawn either in a "brackets" or "radiation" style), as requested already in the first review. The style analogous to the tree included in the Response to Reviewers would be a good start but it needs to be presented as unrooted (because there is no easily identifiable outgroup) and some measure of statistical stability needs to be shown especially for the major branches. I am also puzzled by your statement in the cover letter that the phylogeny was reanalyzed in MEGA but the tree shown was constructed using IQTREE, which is a different software package! Even worse, the trees presented in the manuscript and in the cover letter are not mutually entirely consistent (compare, e.g., the position of At1g19340 or human METTL3). So, how was it really done, and what is going on here?? I suspect that some of the branches are poorly supported and should not be taken too seriously (perhaps all that we can say is that there are the three clades in dicot plants, with no clear relationship to metazoan homologs, and that would be fine for me). I suspaect that based on these data it may be impossible to infer anything about the order of duplication events (besides stating that within-clade duplications are fairly recent), and that the text on l. 177-186 may need some modifications. When preparing the new version of Figure 2, please make sure that ALL lettering is legible, at least 8 pt size (this includes also any statistical support indicators at branches!).

On the occassion of the revision, please clear up also l.108  - the paragraph starts with an isolated reference.

Round 3

Reviewer 1 Report

Excellent work and good analysis.